# Novel Combination of the Biophysical, Nutritional, and Nutraceutical Properties in Subtropical Pigmented Maize Hybrids

**DOI:** 10.3390/plants11233221

**Published:** 2022-11-24

**Authors:** Axel Tiessen-Favier, Anayansi Escalante-Aburto, Claudia Espinosa-Leal, Silverio García-Lara

**Affiliations:** 1Departamento de Ingenieria Genética, Cinvestav, Irapuato 36824, Mexico; 2Tecnológico de Monterrey, Insituto de Obesidad, Monterrey 64849, Mexico; 3Tecnológico de Monterrey, Escuela de Ingenieria y Ciencias, Eugenio Garza Sada 2501, Monterrey 64849, Mexico

**Keywords:** anthocyanin, carotenoids, nutraceutical properties, phenolic acid, pigmented maize, *Zea mays* L.

## Abstract

Maize (*Zea mays* L.) represents the main caloric source for much of the world’s population. Pigmented maize varieties are an excellent source of nutraceutical compounds: blue and yellow maize are rich in anthocyanins as well as carotenoids and phenolic acids, respectively. However, blue maize is usually grown in small quantities as a specialty crop because it lacks the qualities and adaptations of commercial white and yellow varieties. Here, a new high-yield variety of blue maize called Vitamaiz was developed from inbred lines of subtropical blue, white, and yellow maize. The aim of this study was to characterize the nutraceutical and physical properties of 30 Vitamaiz hybrids in two subtropical locations. Kernel physical traits, nutrient composition, and nutraceutical components (free phenolic acids, FPA; cell wall-bound phenolic acids, BPA; total monomeric anthocyanin content, TAC; antioxidant capacity, AOX by oxygen radical absorbance capacity assay, and total carotenoid content, TCC) were evaluated. The biophysical traits of the hybrids were suitable for nixtamalized and flour maize industries. High levels of FPA (228 mg GAE/100 g), BPA (635 mg GAE/100 g), and AOX (2.0 and 8.1 mM Trolox equivalent/100 g for FPA and BPA, respectively) were also detected with elevated TAC levels (274 mg C3G/kg dw) and AOX activity (3.1 mM Trolox equivalent/100 g). This is the first study to characterize Blue × Yellow maize hybrids that adapt to subtropical environments.

## 1. Introduction

Maize is a popular cereal crop that feeds people worldwide [1]; it provides more than 65% of the caloric intake of the poorest population in Mexico as the primary source of protein, vitamins, minerals, lipids, and phytochemicals [2,3]. Before colonization, native American cultures grew a diverse array of pigmented maize genotypes whose colors range from white to black with various intermediate shades of yellow, red, and blue [4,5]. Different types and levels of secondary metabolites, such as phenolic acids, carotenoids, and flavonoids, accumulate in maize and cause pigmentation. These pigments mostly accumulate in the pericarp and aleurone layers of kernels but can also be found in the cobs, husks, and silk [4].

Anthocyanins are natural purple, blue, and red pigments that belong to a major subclass of polyphenols and flavonoids, such as cyaniding-, pelargonidin-, and peonidin-based glucosides, which saturate blue maize with its characteristic color [6]. Flavonoids serve the plant as pollinator recruiters and protect tissues against UV radiation and oxidative stress [4]. Anthocyanins have garnered attention as antioxidants and, therefore, a functional ingredient in maize products [6,7]. Blue maize is traditionally dry-milled and offers a unique flavor in foods such as chips, tortillas, cookies, atoles, pozole, and pinoles [8].

Yellow and orange maize varieties, on the other hand, have high levels of carotenoids, tetraterpenoids, lutein, zeaxanthin, β-carotene, β-cryptoxanthin, and α-carotene [9]. Carotenoids not only enrich plants with rich shades of yellow, orange, and red but also offer health benefits: reduced risk of cancer, cardiovascular diseases, and age-related macular degeneration, enhanced immune responses, and maintenance of healthy skin as well as the gastrointestinal and respiratory systems [10]. Yellow maize is used to produce different products, including gruels, porridge, and infant food [8].

Maize pigmentation is determined not only by plant genotype but also by environmental factors such as altitude, temperature, and abiotic stress. Indeed, it is proposed that the popularity of pigmented varieties declined upon import to Europe because white grains adapted better in Europe [4]. Fortunately, in the last century, the United States and some Latin American countries have reintroduced traditional pigmented maize in their diets [11].

Increasingly productive pigmented hybrids have emerged by adapting maize to diverse agroecologies [9,12]. Traditional breeding is also currently used to promote the biofortification of carotenoids in maize [13]. However, few pigmented hybrids are commercially available in temperate [14,15] and subtropical regions [16]. White and yellow maize varieties adapted to tropical environments have already been selected for high nutraceutical and carotenoid concentrations [17]. Few studies have characterized new blue maize hybrids combined with white and yellow tropical germplasm.

Vitamaiz is a collection of Mexican maize hybrids developed through conventional breeding. It was created to produce high-yield blue–yellow and blue–white maize hybrids that retain most of the properties of pigmented grains, such as phenolic acids, anthocyanin, and carotenes. The aim of this study was to evaluate the physical traits, nutrient composition, and nutraceutical compounds found in 30 promising Vitamaiz hybrids that were developed and tested in two subtropical regions. Furthermore, these compounds and their antioxidant capacities were correlated with the physical properties of the kernels of Vitamaiz hybrids.

## 2. Results

### 2.1. Biophysical Properties

The properties of 30 Vitamaiz hybrids were evaluated in two subtropical environments in 2018 (Table 1). The main physical traits of the hybrids ranged from 5.6–9.0 Ton/ha for yield, 76–82 kg/hL for test weight, 269–400 g for thousand kernel weight (TKW), and 0.5–34.7% for flotation index (FI). Significant differences (*p* < 0.001) in all physical traits evaluated were found between the genotypes; however, the environment had no significant effects. The heritability was high (>0.7) only for yield, test weight, and TKW. Differences between the types of crosses (Blue × White vs. Blue × Yellow) were significant only for test weight, TKW, color (b), and kernel thickness. Overall, the Vitamaiz hybrids presented biophysical traits suitable for nixtamalized and flour maize industries.

### 2.2. Nutritional Properties

Table 2 shows the results of the combined analysis of the nutritional parameters of Vitamaiz hybrids grown in two subtropical environments. Based on the proximal analysis, the major components were proteins (9.1–10.7%), oil (3.5–4.8%), and fibers (0.6–1.6%). Significant differences (*p* < 0.001) in genotype were observed for all nutrimental traits evaluated; however, the environment had no significant influence. The heritability was low (<0.5) for proteins, oil, and fiber. Significant differences between the types of crosses were also found for all nutrimental traits; Blue × White had higher nutritional content than Blue × Yellow. Generally, Vitamaiz hybrids contained a high nutritional value for animal or human consumption.

### 2.3. Nutraceutical Properties

Figure 1 describes the core nutraceutical properties of kernels from 30 Vitamaiz hybrids. Specifically, phenolic acid content (total soluble and cell wall-bound phenolic acids), total anthocyanins and their associated antioxidant capacity, and total carotenoid content are presented as results of the combined analysis of the data from two subtropical environments. Significant differences (*p* < 0.001) in all nutraceutical traits based on genotype and genotype × environment (G × E) were observed. No environmental effects were detected for free phenolic acids, AOX of anthocyanins, and carotenoid content. The heritability was low (<0.2) for all nutraceutical traits except for carotenoid content, which presented a high heritability value of 0.88.

### 2.4. Free and Bound Phenolic Acids and Antioxidant Capacity

The total soluble and cell wall-bound phenolic acid contents ranged from 125–228 to 359–635 mg GAE/100 g dw, respectively. The antioxidant capacity of soluble and cell wall-bound phenolic acids ranged from 1172–2000 to 4346–8280 μmol TE/100 g dw, respectively. Vm492 × VmM451 (Blue × Yellow) and MzDtp413 × VmM451 (Blue × Yellow) had the highest soluble and bound phenolic contents, respectively. Furthermore, Vm451 × Vm311 (Blue × Yellow) and Vm492 × VmM451 (Blue × Yellow) achieved the highest antioxidant activity for soluble and bound phenolic acid, respectively.

### 2.5. Anthocyanin Content and Antioxidant Capacity

The total anthocyanin content and antioxidant capacity ranged from 114–274 mg C3G/kg dw to 1737–3111 μmol TE/100 g dw, respectively. Interestingly, Vm245 × Vm311 (Blue × White) had the highest total anthocyanin content, while Vm451 × Vm311 (Blue × Yellow) showed the highest OAX activity.

### 2.6. Total Carotenoid Content of Vitamaiz Hybrids

The total carotenoid content of the Vitamaiz hybrids was also measured; it ranged from 0.46 to 8.28 µg of bCE/100 g dw; Vm327 × Vm451 (Blue × Yellow) had the highest total carotenoid content. From these hybrids, the parent CML451 expressed the hybrid combination whose concentration of nutraceutical parameters was highest.

### 2.7. Comparison between Crosses

The two types of crosses, Blue × White and Blue × Yellow, were further compared, as shown in Figure 1. The free phenolic acid content of Blue × White was significantly greater than Blue × Yellow. Furthermore, the Blue × White cross had the greatest ORAC activity, while the Blue × Yellow cross had the highest total carotenoid content. 

### 2.8. Association Analysis

A correlation analysis was performed to determine whether nutraceuticals were associated with the physical and nutritional properties of the Vitamaiz hybrids (Table 3). Statistically significant and biologically relevant correlations were found. First, FPA content was positively correlated with ash content, TW, TKW, and kernel length but negatively with FI. In contrast, BPA content was negatively correlated with calculated total carbohydrate content (CHO) and ash content and positively correlated with fiber content. Interestingly, the AOX activity of FPA, BPA, and anthocyanins was positively correlated with fiber content, KWK, and kernel length. Finally, carotenoid content was associated with fiber, protein content, and FI and was negatively correlated with oil, CHOs, and TKW. 

## 3. Discussion

### 3.1. Physical Properties of Novel Hybrids

As a novel source of nutraceutical compounds, the Vitamaiz hybrids are desirable not only for their physical and nutritional properties; their yield is competitive (7–9 Ton/ha) in local agricultural practices relative to commercially available, conventional blue hybrids grown in subtropical environments [18] and to other, specialty conventional pigmented maize hybrids [19].

The physical characteristics of kernels affect their performance in processing and the acceptability of generated products [20]. The test weight of the Vitamaiz hybrids was consistent with Uriarte-Aceves et al. [21], who measured the test weights of 15 Mexican blue maize genotypes from 71 to 84 kg/hL. The Vitamaiz hybrid grains ranged from what was established for nixtamalized products [22]. A test weight of 76–78 kg/hL and a TKW of 290–340 g are ideal for tortilla production [20]; five hybrids (ID 11, 18, 24, 28, and 30) in this study fulfilled these criteria. The ideal endosperm hardness depends on the end product; kernels with hard endosperms (FI ≤ 20%) are preferred for nixtamalized flour, while a softer endosperm (FI ≤ 40%) is suitable for masa and tortilla [23]. The Vitamaiz hybrids had the desired TW and TKW, and their endosperm hardness renders them suitable for both nixtamalized flour and masa and tortilla production.

### 3.2. Nutraceuticals in Pigmented Maize

The total soluble phenolic acid content of the Vitamaiz hybrids was higher than that of blue kernels evaluated by Mora-Rochin et al. [24] and in the upper range of blue kernels hybrids studied by Urias-Lugo et al. [12]. The AOX associated with the phenolic acid content was lower than that reported by Mora-Rochin et al. [24], but it was within the range measured by Urias-Lugo et al. [12]. The level of cell wall-bound phenolic acids and the antioxidant capacity of the Vitamaiz hybrids were higher than those reported by Mora-Rochin et al. [24] in blue kernels and Urias-Peraldí et al. [25] in white–blue kernels hybrids. In this study, the bound phenolics content was within the range reported by López-Martínez et al. [26] in pigmented kernels with purple, black, and red colors. Mora-Rochin et al. [24] also measured the AOX of pigmented kernels and found that white, yellow, and red had a higher capacity than blue kernels. The cross between blue and yellow or white kernels modified the phenolic profiles of the resulting hybrids, which could explain the higher AOX capacities of the Vitamaiz hybrids [27]. 

Overall, Vitamaiz hybrids had low anthocyanin content and AOX compared to previous reports. For example, the ranges of anthocyanin content found here were lower than those of blue kernels studied by Mora-Rochin et al. [24] and López-Martínez et al. [26], white–blue kernels hybrids studied by Urias-Peraldí et al. [25], and other pigmented kernels, including purple, black, red, and pink studied by López-Martínez et al. [26]; it was only within the range measured by Preciado-Ortiz et al. [19] for red and purple specialty varieties. Anthocyanin accumulation and metabolism depend on several environmental factors, which were also detected in this study (significant interaction G*E) and previously reported for these metabolites [25,28]. 

The genetic profile of Vitamaiz could hinder anthocyanin accumulation because of the combinatory capacity observed previously [29]. Although the anthocyanin content of Vitamaiz hybrids was lower than that of other blue hybrids, their high total phenolic content and associated AOX capacity indicate their functional potential. The carotenoid content of the Vitamaiz hybrids fell within the range reported by Kuhnen et al. [30] for yellow and purple Brazilian maize landraces. Maize genotype affects the carotenoid concentration [31,32]; Vitamaiz hybrids from the parental line CML451 may have a greater ability to accumulate these compounds [33].

### 3.3. Effects of Genetic Background and Origin

Two types of crosses were compared in this study: Blue × White was superior to Blue × Yellow in terms of nutraceutical content and activity, excluding carotenoids. This genetic advantage has been previously reported for similar crosses with white maize [34]. This is not surprising because they originated from different ecoregions, and these factors reflect how these hybrids adapt to various cultivation environments, stressors, and endosperm characteristics [35]. Phenolic compounds regulate plant physiology, adaptation, and resistance; for example, phenolic compounds accumulate in plants subjected to mechanical and biological stresses [36]. In this study, new hybrids from both crosses were excellent sources of nutraceuticals.

### 3.4. Physical, Nutrimental, and Nutraceutical Associations

The Vitamaiz hybrids contained compounds (soluble and bound phenolics, anthocyanins, and carotenoids) that were related to the fiber content and other physical characteristics of the kernels; carotenoids were specifically linked to oil as well as fiber. Maize fiber primarily contains the kernel outer seed coat (pericarp) [37] because the metabolites studied are concentrated in the kernel pericarp [4]. Marques et al. [37] found that maize fiber is a rich source of lipophilic compounds, a characteristic shared by carotenoids; however, they did not measure the total carotenoid content of the fiber. This suggests that fiber content contributes to the accumulation of nutraceutical compounds and the AOX capacity [26,38,39], which may also contribute to the physical properties of maize grains.

Therefore, Vitamaiz hybrids represent an opportunity to expand the use of pigmented maize to other geographical zones, such as subtropical areas. The grains from these hybrids also contained high levels of nutraceuticals, especially phenolic acid compounds that could be used to produce tortillas and other regional pigmented maize products.

## 4. Materials and Methods

### 4.1. Maize Germplasm and Hybrid Production

Improved subtropical blue, yellow, and white maize inbred lines were combined to develop a new high-yield generation of subtropical maize hybrids collectively named Vitamaiz (*Zea mays* L. sp. mays var. Vitamaiz). These experimental hybrids of Vitamaiz were developed by the National Laboratory PlanTECC CINVESTAV-Irapuato, as previously described [40]. Nurseries and field trials were performed as previously described [41]. The breeding process included allele introgression through repetitive backcrossing to recurrent parents (blue parent combined with selected tropical CML, Blue × CML-White, and Blue × CML-Yellow) coupled with intensive selection based on biochemical data and phenotype traits. The resulting crosses, Blue × White or Blue × Yellow, were selected for a subsample from 30 hybrids (15 Blue × White & 15 Blue × Yellow) that were obtained from superior crosses planted during the summer of 2018 in west-central Mexico in two subtropical environments (semiwarm humid with summer rains, Aw1) according to the INEGI [42], the Celaya Guanajuato, (N 20°31′44″, W 100°48′54″; 1750 m above the sea level, masl) and Rancho Santa Margarita, Puerto Vallarta, Jalisco (N 20°44′, W 105°10′; 731 masl). Both experimental plots had two rows of 5.0 m length, 0.76 m of separation, and three reps per hybrid; the experiment was fertilized twice with a total of 250 U of N, 60 U of P, and 30 U of K per hectare. Irrigation, pest management, and weed control were performed using standard practices in the region. Sowing and harvesting were performed manually. The ears were manually harvested, dried, and shelled. A total of 64 kernel samples from each field replica were obtained to form a representative pool of grains. Then, 1000 g of each sample was sent to a laboratory for seed analysis and immediately placed in plastic-sealed containers and stored at −4 °C until further use.

### 4.2. Measurement of Biophysical Properties

The physical properties of the grains were measured using standard procedures: test weight according to Official US Grain Standard Procedures [43]; TKW by weighing 1000 randomly selected kernels at 13% of grain moisture [44]; FI according to García-Lara et al. [36]; and grain hardness as the percentage of floating kernels on an aqueous sodium nitrate solution with a specific weight of 1.25 g/cm^3^ at 35 °C. The samples were ground using a grinder (mesh sieves no. 16) equipped with a 0.5-HP motor (Krups GX 4100-11, Medford, MA, USA). The ground samples were placed in 20 mL vials and stored at −20 °C for nutraceutical and physical analyses. Color parameters L*, a*, and b* of the ground samples were determined using a colorimeter (Minolta CR-300, Osaka, Japan).

### 4.3. Nutrimental Analysis

Near-infrared spectroscopy (NIR) was used as a noninvasive method for chemical analysis of the different types of genotypes. Proximate composition was performed with a NIR spectrometer equipped with a diode-array light source (DA7250, Perten, Springfield, IL, USA) and a Perten calibration package SimPlus (Software v.2009, Springfield, IL, USA). Protein, oil, and starch were determined and expressed as dry matter. Bio-based corrections for NIR data were performed according to the manufacturer’s instructions.

### 4.4. Quantification of Total Soluble and Bound Phenolic Acids

Ground samples were used for the microextraction of soluble and bound phenolic compounds, as described previously [45]. Briefly, 80% methanol was used to extract soluble phenolic compounds in triplicate for each hybrid. The remaining pellets were subjected to alkaline hydrolysis for 1 h and neutralized. Two hexane washes were used to remove lipids, and three ethyl-acetate washes were used to recover cell wall-bound phenolic compounds. Bound phenolic extracts were dried with nitrogen and dissolved in 50% methanol. All extracts were stored at −20 °C until further analysis. Soluble and bound phenolic acids were quantified using the Folin-Ciocalteau assay [46]. The samples were first oxidized using a Folin-Ciocalteau reagent, mixed for 3 min, and then neutralized with sodium bicarbonate. Next, the solution was mixed for 7 min with 400 µL of distilled water. After 1 h of further mixing, the solution was read at 726 nm using a microplate reader (Synergy HTX Multi-Mode Reader, BioTek, Santa Clara, CA, USA). The experimental samples were compared to a gallic acid standard curve prepared from seven different gallic acid solutions whose concentrations ranged from 50 ppm to 200 ppm in 25 ppm increments; the resulting R^2^ was 0.99. Data are expressed as mg of gallic acid equivalents per 100 g of dry weight (mg of GAE/100 g dw).

### 4.5. Total Anthocyanin Quantification

Total anthocyanins were assayed according to Abdel-Aal et al. [47] with some modifications. First, 600 mg of ground samples was weighed in a 15 mL centrifuge tube, and 4.8 mL of acidified ethanol was added (ethanol + HCl 1 N, 85:15). The mix was agitated in a vortex for 30 min; then, the pH was adjusted to 1 with 4N HCl if necessary. The tubes were centrifuged at 3000 rpm for 10 min, and the supernatant was recovered. Absorbance was measured at 520 nm using a UV–Vis spectrophotometer (Thermo Scientific, Model Evolution 300, Austin, TX, USA). Cyanidin 3-glucoside or kuromanin (Sigma-Aldrich) was used as a standard pigment. A series of cyanidin 3-glucoside standard solutions was prepared at 0–0.02 mmol (0–27 µg/3 mL). Data are expressed as mg of cyanidin-3 glucoside equivalents per kg of dry weight (mg C3G/kg dw).

### 4.6. Antioxidant Capacity Determination

Antioxidant capacity was determined using the ORAC assay. Extracts from soluble and bound phenolic acids and anthocyanins were evaluated against a standard of Trolox with fluorescein as a probe, as described by Ou et al. [48]. Briefly, samples were diluted in 75 mM pH 7.4 phosphate-buffered saline (PBS). A radical generator was prepared using 153 mM 2,2′-Azobis (2-amidinopropane) dihydrochloride (AAPH) (Sigma-Aldrich, St. Louis, MO, USA) and dissolved in PBS. Excitation/emission was measured at 485/520 nm at 37 °C using a microplate reader (Synergy HTX Multi-Mode Reader, BioTek, Santa Clara, CA, USA). A standard curve was prepared using Trolox solutions with 20, 40, 60, 80, and 100 µM concentrations; the R^2^ was 0.99. Data are expressed as µmol of Trolox equivalents/100 g of dry sample weight (µmol TE/100 g dw).

### 4.7. Total Carotenoid Content Determination

Total carotenoid content was measured as described by Kurilich et al. [32]. Ground samples (60 mg) were mixed with 600 µL of 0.1% terbutilhidroquinone (TBHQ; Sigma-Aldrich, St. Louis, MO, USA) for saponification. Then, the samples were incubated at 85 °C for 5 min with constant agitation at 500 rpm. An 80% KOH (50 µL) solution was added, and the mix was agitated. The samples were incubated again at 85 °C for 10 min with constant agitation at 500 rpm. Bi-distilled water (300 µL) was added, and the samples were immediately placed on ice. Then, 300 µL of hexane was added, and the samples were centrifuged for 10 min at 1300× *g* to recover the supernatant. This step was repeated twice to complete three hexane washes. The collected superior phases were washed with 300 µL of water and centrifuged for 10 min at 1200 rpm. The superior phase was measured at 450 nm (Synergy HTX Multi-Mode Reader, BioTek, Santa Clara, CA, USA). The experimental samples were compared to a beta-carotene standard curve. Data are expressed as µg of beta-carotene equivalents per 100 g of dry weight (µg of bCE/100 g dw).

### 4.8. Statistical Analysis

Results are expressed as mean ± standard error of three replicates. Analysis of variance (ANOVA) was performed to detect differences between the samples, followed by multiple comparisons with Fisher’s least significant difference (LSD) to identify specific differences between the types of hybrids and interactions between environments (sites). Broad heritability values were calculated according to Hallauer et al. [49]. ANOVA, multiple comparisons, and Pearson correlations were analyzed using the statistical software Statistix v.7 (Analytical Software, Tallahassee, FL, USA). A *p*-value of 0.05 or less was considered significant.

## 5. Conclusions

In this study, the biophysical, nutritional, and nutraceutical properties of 30 pigmented hybrids developed for subtropical locations were evaluated. New hybrids from crosses of blue × white and blue × yellow maize, especially the parental line CML451, proved to be excellent sources of combined nutraceuticals because of their greater potential to accumulate compounds such as phenolic acid. The high antioxidant capacity and endosperm hardness of Vitamaiz hybrids render them suitable for the production of nixtamalized flour, masa, and tortillas, as well as traditional, local maize products. Vitamaiz hybrids represent an opportunity to expand the use of pigmented maize to other geographical zones, such as subtropical areas, to increase food availability and sustainability. Nevertheless, future studies should enhance the genetic variation of naturally pigmented maize to improve its nutraceutical profile.

## Figures and Tables

**Figure 1 plants-11-03221-f001:**
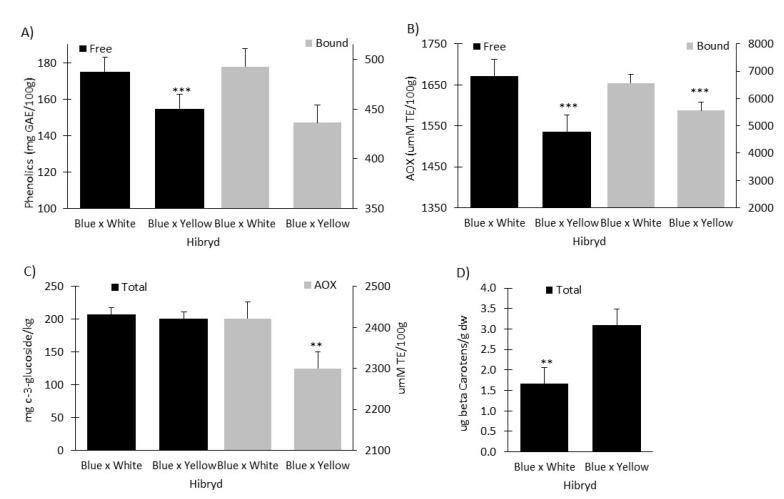
Mean comparative analysis of nutraceutical compounds and antioxidant activity of kernels from 30 subtropical blue maize hybrids “Vitamaiz” divided by its crosses. (**A**) Free and bound phenolic acid content. (**B**) Antioxidant activity of free and bound phenolic acids. (**C**) Anthocyanin content and antioxidant activity. (**D**) Total carotenoid content. ** Significant at *p* < 0.01. *** Significant at *p* < 0.001; Bars on each sample mean the standard error of three replicates.

**Table 1 plants-11-03221-t001:** Combined analysis of the yield and kernel properties of 30 Vitamaiz hybrids grown in two subtropical environments (CEL—Celaya; PV—Puerto Vallarta, Mexico) in 2018.

ID	Hybrid Cross	Genetic Source BC Parent	Yield Ton ha^−1^	Test Weight kg/hL	TKW ^1^ g	FI %	Color		Kernel Dimensions (mm)		
							b	L	Thickness	Width	Length
1	Vm451 × Vm311	CML451 (Y)	8.3	78.5	400.1	3.0	10.9	42.6	3.5	9.1	12.6
2	Vm496 × Vm311	CML496 (Y)	7.9	79.1	330.9	4.8	10.4	36.1	3.4	8.7	11.5
3	Vm496 × Vm451	CML451 (Y)	8.1	79.8	328.3	1.7	13.5	51.7	3.0	9.0	11.1
4	Vm492 × Vm451	CML451 (Y)	7.5	82.0	355.2	5.7	7.7	38.8	4.2	8.7	10.5
5	MzDtp222 × Vm311	CML311 (W)	6.9	79.3	274.5	3.5	13.1	56.1	2.8	8.5	10.6
6	MzDtp222 × Vm321	CML321 (W)	5.6	78.9	269.7	8.5	10.4	49.3	3.1	8.7	10.4
7	MzDtp222 × Vm451	CML451 (Y)	8.2	80.0	336.7	4.3	13.7	47.0	3.2	8.7	11.4
8	MzDtp411 × Vm311	CML311 (W)	6.7	80.4	333.3	2.2	10.5	47.7	3.4	8.9	11.1
9	MzDtp411 × Vm321	CML321 (W)	6.0	78.3	302.8	11.3	10.7	45.4	3.1	9.2	10.7
10	MzDtp413 × Vm451	CML451 (Y)	8.4	80.0	342.8	4.2	11.3	41.3	3.4	9.0	11.2
11	MzDtp831 × Vm321	CML321 (W)	7.3	77.3	321.8	16.5	6.5	39.4	3.6	9.5	11.1
12	MzDtp831 × Vm451	CML451 (Y)	8.2	78.8	347.4	18.0	12.3	47.1	3.8	9.0	11.3
13	MzDtpY1212 × Vm451	CML451 (Y)	9.2	79.6	403.7	3.8	14.4	49.8	3.8	9.5	11.9
14	MzLPSF86 × Vm311	CML449 (W)	7.2	78.5	340.5	10.5	9.1	41.0	3.0	9.5	11.3
15	MzLPSF86 × Vm321	CML321 (W)	7.1	78.5	304.8	7.2	11.0	47.2	2.5	9.2	10.7
16	MzLPSF86 × Vm451	CML451 (Y)	8.1	80.3	350.2	3.3	9.1	39.7	3.5	8.8	11.5
17	VmLPSF103 × Vm451	CML451 (Y)	7.4	82.3	313.1	0.5	6.1	32.7	3.1	8.1	11.3
18	Vm254 × Vm311	CML311 (W)	6.6	77.7	300.6	9.0	11.6	53.0	2.8	8.9	11.2
19	Vm254 × Vm321	CML321 (W)	6.0	78.8	310.9	6.2	6.9	39.5	2.9	9.2	11.2
20	Vm321 × Vm311	CML321 (W)	7.1	78.3	288.9	7.5	12.0	49.2	3.3	8.9	10.7
21	Vm492 × Vm311	CML311 (W)	6.0	79.2	310.7	10.8	10.7	44.6	3.3	8.6	10.6
22	Vm492 × Vm321	CML321 (W)	7.3	77.4	282.3	21.5	9.1	42.8	2.8	8.8	10.5
23	Vm492 × Vm451	CML451 (Y)	7.2	80.9	372.7	5.3	8.5	37.1	4.3	8.5	11.3
24	Vm311 × Vm321	CML311 (W)	6.4	76.4	299.6	17.0	7.6	39.9	3.0	8.9	10.8
25	Vm311 × Vm451	CML451 (Y)	7.3	79.1	334.3	13.8	14.0	50.9	4.1	8.8	10.4
26	Vm490 × Vm311	CML311 (W)	6.4	80.4	312.0	2.3	10.0	47.7	3.4	8.9	10.8
27	Vm490 × Vm321	CML321 (W)	6.6	79.5	300.6	7.0	7.6	38.1	3.1	9.1	10.5
28	Vm327 × Vm321	CML496 (Y)	7.1	76.1	309.7	34.7	12.1	43.5	3.1	9.0	10.3
29	Vm327 × Vm451	CML451 (Y)	9.0	79.8	363.0	8.3	12.8	38.4	3.8	8.4	11.1
30	Vm338 × Vm311	CML338 (Y)	7.3	75.5	318.7	16.3	15.7	50.3	3.0	9.4	11.2
	Mean by Cross (C)	Blue × White	6.6	79.4	347.1	8.5	11.5	43.1	3.54	8.9	11.2
		Blue × Yellow	7.9	78.5	303.5	9.4	9.8	45.4	3.07	9.0	10.8
	Cross (C)		ns	**	***	ns	**	ns	***	ns	ns
	Genotype (G)		***	***	***	***	***	***	***	***	***
	Genotype LSD (0.05)		2.1	0.3	7.9	3.4	4.1	10.2	0.2	0.3	0.4
	Environment (E)		ns	ns	ns	ns	ns	ns	ns	ns	ns
	G×E		ns	***	***	*	***	***	***	ns	ns
	Heritability		0.71	0.76	0.88	0.45	0.01	0.01	0.26	0.12	0.16

* Significant at *p* < 0.05. ** Significant at *p* < 0.01. *** Significant at *p* < 0.001; ns—not significant. ^1^ TKW—thousand kernel weight; FI—flotation index; b and L are color properties.

**Table 2 plants-11-03221-t002:** Combined analysis of the nutritional composition of kernels from 30 Vitamaiz hybrids grown in two subtropical environments (CEL—Celaya; PV—Puerto Vallarta, Mexico) in 2018.

ID	Hybrid Cross	Genetic Source BC Parent	Protein	Oil	CHOs ^1^	Fiber	Ash
			g/100 g of Dry Weight				
1	Vm451 × Vm311	CML451 (Y)	10.0	3.9	60.3	1.3	1.1
2	Vm496 × Vm311	CML496 (Y)	9.4	4.2	61.3	0.9	1.1
3	Vm496 × Vm451	CML451 (Y)	9.7	4.4	59.7	1.1	1.0
4	Vm492 × Vm451	CML451 (Y)	10.4	4.0	59.6	1.4	1.1
5	MzDtp222 × Vm311	CML311 (W)	10.3	4.2	61.2	0.6	1.1
6	MzDtp222 × Vm321	CML321 (W)	9.6	4.3	61.1	0.9	1.0
7	MzDtp222 × Vm451	CML451 (Y)	9.4	4.3	60.2	1.1	1.0
8	MzDtp411 × Vm311	CML311 (W)	9.8	4.5	60.3	0.8	1.1
9	MzDtp411 × Vm321	CML321 (W)	9.5	4.6	60.3	1.0	1.0
10	MzDtp413 × Vm451	CML451 (Y)	10.1	4.2	59.1	1.4	1.0
11	MzDtp831 × Vm321	CML321 (W)	9.4	4.4	60.4	1.0	1.0
12	MzDtp831 × Vm451	CML451 (Y)	10.0	4.3	59.7	1.3	1.1
13	MzDtpY1212 × Vm451	CML451 (Y)	10.5	3.9	59.4	1.3	1.1
14	MzLPSF86 × Vm311	CML449 (W)	9.4	4.3	61.6	0.8	1.1
15	MzLPSF86 × Vm321	CML321 (W)	9.6	4.5	60.1	1.0	1.0
16	MzLPSF86 × Vm451	CML451 (Y)	10.0	4.1	60.1	1.1	1.1
17	VmLPSF103 × Vm451	CML451 (Y)	9.4	4.2	60.6	1.0	1.1
18	Vm254 × Vm311	CML311 (W)	9.0	4.1	61.5	1.1	1.1
19	Vm254 × Vm321	CML321 (W)	9.3	4.4	60.8	1.0	1.1
20	Vm321 × Vm311	CML321 (W)	9.6	4.3	61.0	1.0	1.0
21	Vm492 × Vm311	CML311 (W)	9.6	4.4	61.7	0.6	1.1
22	Vm492 × Vm321	CML321 (W)	9.8	4.5	61.0	0.7	1.1
23	Vm492 × Vm451	CML451 (Y)	9.9	4.1	60.6	1.0	1.1
24	Vm311 × Vm321	CML311 (W)	9.4	4.1	62.0	0.9	1.0
25	Vm311 × Vm451	CML451 (Y)	10.3	4.0	60.4	1.2	1.1
26	Vm490 × Vm311	CML311 (W)	9.2	4.8	60.4	0.8	1.1
27	Vm490 × Vm321	CML321 (W)	9.1	4.8	60.4	0.8	1.1
28	Vm327 × Vm321	CML496 (Y)	9.9	3.5	60.4	1.6	1.0
29	Vm327 × Vm451	CML451 (Y)	10.7	3.7	59.5	1.4	1.1
30	Vm338 × Vm311	CML338 (Y)	10.0	4.6	60.1	1.0	1.0
	Cross Mean	Blue × White	10.0	4.1	60.1	1.2	1.1
		Blue × Yellow	9.5	4.4	60.9	0.9	1.1
	Cross (C)		***	***	***	***	ns
	Genotype (G)		***	***	***	***	***
	Genotype LSD (0.05)		2.1		0.3	7.9	4.1
	Environment (E)		ns	ns	ns	ns	ns
	G × E		**	ns	ns	**	ns
	Heritability		0.42	0.09	0.27	0.25	0.24

** Significant at *p* < 0.01. *** Significant at *p* < 0.001; ns—not significant. ^1^ CHOs—calculated total carbohydrate content.

**Table 3 plants-11-03221-t003:** Pearson correlation analysis between biochemical, biophysical, and nutraceutical properties of kernels from 30 Vitamaiz hybrids grown in two subtropical environments in 2018.

Trait	FPA		BPA		Nutraceutical							
					AOX-FPA		AOX-BPA		AOX-Ant		Carotenoids	
Biochemical												
Oil											−0.311	**
Protein											0.403	***
CHOs			−0.547	***	−0.260	*	−0.526	***	−0.260	*	−0.255	*
Fiber			0.422	***	0.262	*	0.448	***	0.390	**	0.355	**
Ash	0.372	***	−0.443	***								
Physical												
TW	0.480	***			0.387	**	0.447	***	0.432	***		
TKW	0.395	**					0.358	**			−0.316	**
FI	−0.277	*			−0.291	*	−0.340	**			0.388	**
Length	0.415	***			0.358	**	0.366	**	0.488	***		

* Significant at *p* < 0.05. ** Significant at *p* < 0.01. *** Significant at *p* < 0.001; AOX—antioxidant; FPA—free phenolic acids; BPA—bound phenolic acids; AOX—antioxidant activity; ANT—anthocyanins; CHOs—calculated total carbohydrate content; TKW—thousand kernel weight; TW—test weight; FI—flotation index.

## Data Availability

Data Availability only under request.

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
