# Peer review of "Novel Combination of the Biophysical, Nutritional, and Nutraceutical Properties in Subtropical Pigmented Maize Hybrids"

_plants, 2022, doi:10.3390/plants11233221_

Round 1
Reviewer 1 Report
This manuscript titled "Novel Combination of Nutraceutical Properties in Pigmented Maize Hybrids (Zea mays)" provides valuable information on properties of pigmented maize hybrids adapted to subtropical environment.
The introduction provides a generalized but detailed background of the topic and explains the objective of the paper.
The results are well explained. However, there are some oversights in the presentation of the results. Specific comments are listed below.
Specific comments:
page 2, lines 75-76: Rewrite the sentence by omitting the redundant part.
page 2, lines 77-78: Match the yield range and TKW data with the data shown in Table 1.
page 2, lines 85-94: Table 1. is cited in paragraphs 2.1. and 2.2. However, according to the results shown in the Table 1, it seems that the table cited in the paragraph 2.2. is missing in the manuscript.
page3, line 102: Match the protein content data with the data shown in the Table 2.
page 5, line 125: Correct the antioxidant capacity abbreviation.
page 6, line 146-147: Check the correlation between FPA and FI. Positive correlation is stated in the text, while in the Table 3. a negative correlation is shown.
page 6, lines 147 and 151: It seems that the abbreviation KWK in paragraph 2.8. stands for test weight (TW). Please check and revise it.
page 6, Table 3.: In the first column of the Table 3., starch is listed under biochemical properties. Revise and match what is shown in the Table with the explanation below the Table and in paragraph 2.8.
Author Response
This manuscript titled "Novel Combination of Nutraceutical Properties in Pigmented Maize Hybrids (Zea mays)" provides valuable information on properties of pigmented maize hybrids adapted to subtropical environment. The introduction provides a generalized but detailed background of the topic and explains the objective of the paper. The results are well explained. However, there are some oversights in the presentation of the results. Specific comments are listed below.
Specific comments:
page 2, lines 75-76: Rewrite the sentence by omitting the redundant part.
R= The sentence was changed according to the reviewer's suggestion
page 2, lines 77-78: Match the yield range and TKW data with the data shown in Table 1.
R= The sentence was changed according to the reviewer's suggestion
page 2, lines 85-94: Table 1. is cited in paragraphs 2.1. and 2.2. However, according to the results shown in the Table 1, it seems that the table cited in the paragraph 2.2. is missing in the manuscript.
R= We appreciate these observation. The table was corrected according to the reviewer's suggestion
page3, line 102: Match the protein content data with the data shown in the Table 2.
R= The protein data was changed according to the reviewer's suggestion
page 5, line 125: Correct the antioxidant capacity abbreviation.
R= The sentence was changed according to the reviewer's suggestion
page 6, line 146-147: Check the correlation between FPA and FI. Positive correlation is stated in the text, while in the Table 3. a negative correlation is shown.
R= The sentence was changed according to the reviewer's suggestion
page 6, lines 147 and 151: It seems that the abbreviation KWK in paragraph 2.8. stands for test weight (TW). Please check and revise it.
R= The KWK was changed according to the reviewer's suggestion
page 6, Table 3.: In the first column of the Table 3., starch is listed under biochemical properties. Revise and match what is shown in the Table with the explanation below the Table and in paragraph 2.8.
R= Starch was changed by CHOs according to the reviewer's suggestion
Reviewer 2 Report
The paper is interesting with a current topic. It is written clearly, at a high level, with clearly defined objectives, extensive discussion and properly made conclusions.
The results can find practical application in the food industry (functional food).
Author Response
Reviewer #2:
The paper is interesting with a current topic. It is written clearly, at a high level, with clearly defined objectives, extensive discussion and properly made conclusions.
The results can find practical application in the food industry (functional food).
R= Thank you very much for your comments.
Reviewer 3 Report
Dear Authors,
In my opinion, your manuscript “Novel Combination of Nutraceutical Properties in Pigmented Maize Hybrids (Zea mays)" presented for my assessment is written in the correct form and the information contained in it could be cognitive and application significant after some refilling. Although the work is interesting, I think that the Authors should take count a modification of this article. I recommend publishing it in "Plants" after a minor revision.
General comments:
1. The manuscript needs some revision, especially for language, I feel that editing the text by a native English speaker would help and improve the readability of the paper.
Several specific comments:
Title:
Novel Combination of Nutraceutical Properties in Pigmented Maize Hybrids (Zea mays) – Novel Combination of Nutraceutical Properties in Pigmented Maize Hybrids (without the Latin species name in this place).
Abstract:
Line 24 - 25: “This is the first study to characterize Blue × Yellow maize hybrids that adapt to subtropical environments and could be used to produce tortillas and other maize products”. - in my opinion, the underlined part of this sentence is not of scientific meaning and should be changed.
Keywords:
If the Authors cancel “Zea mays L.” from the title, it can be placed in keywords.
Results
Line 113 – 142: this part of results is not clear, while the description of the results should be clear and unambiguous so that the reader does not guess but can be sure. I could not find the citation of Figure 1 in the text (I understand Figure 1 is correlated with the 2.4, 2.5, 2.6, and 2.7 chapters) and the individual graphs of Figures 1 A, B, C, and D. It should also be explained what the whiskers in the charts mean or the standard deviation or, for example, the confidence intervals?
Line 153: Table 3 – there is no CHOs in the table.
Materials and Methods
Line 235: Zea mays L. – Zea mays L.
Line 238: Briefly, the breeding process ….. – cancel “Briefly,”
I could not find the information about the carbohydrate content estimation
Conclusions
I advise you to emphasize the conclusions more.
With best regards!
Author Response
Reviewer #3:
In my opinion, your manuscript “Novel Combination of Nutraceutical Properties in Pigmented Maize Hybrids (Zea mays)" presented for my assessment is written in the correct form and the information contained in it could be cognitive and application significant after some refilling. Although the work is interesting, I think that the Authors should take count a modification of this article. I recommend publishing it in "Plants" after a minor revision.
General comments:
The manuscript needs some revision, especially for language, I feel that editing the text by a native English speaker would help and improve the readability of the paper.
R= Professional edition service was applied to the manuscript.
Several specific comments:
Title: Novel Combination of Nutraceutical Properties in Pigmented Maize Hybrids (Zea mays) – Novel Combination of Nutraceutical Properties in Pigmented Maize Hybrids (without the Latin species name in this place).
R= We have eliminated the Latin specie as the reviewer requested.
Abstract: Line 24 - 25: “This is the first study to characterize Blue × Yellow maize hybrids that adapt to subtropical environments and could be used to produce tortillas and other maize products”. - in my opinion, the underlined part of this sentence is not of scientific meaning and should be changed.
R= The sentence was changed according to the reviewer's suggestion (line 25).
Keywords: If the Authors cancel “Zea mays L.” from the title, it can be placed in keywords.
R= The keyword “Zea mays L.” was included in lines 26-27.
Results
Line 113 – 142: this part of results is not clear, while the description of the results should be clear and unambiguous so that the reader does not guess but can be sure. I could not find the citation of Figure 1 in the text (I understand Figure 1 is correlated with the 2.4, 2.5, 2.6, and 2.7 chapters) and the individual graphs of Figures 1 A, B, C, and D. It should also be explained what the whiskers in the charts mean or the standard deviation or, for example, the confidence intervals?
R= We have deleted the sentence in lines 143-144 “Both Vitamaiz hybrids were good sources of nutraceuticals and had promising antioxidant capacities.” according to the reviewer suggestion. Also, Figure 1 is cited in line 141 (section 2.7). The legend “ Bars on each sample mean the standard error of three replicates.” was included in Figure 1
Line 153: Table 3 – there is no CHOs in the table.
R= The correction has been made in table 3.
Materials and Methods
Line 235: Zea mays L. – Zea mays L.
R= The correction has been made
Line 238: Briefly, the breeding process ….. – cancel “Briefly,”
R= The word briefly was deleted in line 257.
I could not find the information about the carbohydrate content estimation
R= A section was included describing how the proximal analysis was performed in the samples (section 4.1).
Conclusions
I advise you to emphasize the conclusions more.
R= The conclusion section was changed according to your suggestions.
Reviewer 4 Report
Dear Authors,
the topic of comparing the nutritional and pro-health value (based on the content of bioactive ingredients) presented for review is interesting. At the same time, I would like to point out that the discussion of the results requires supplementing the issue of the content of selected bioactive elements, the influence of the genome, and location. This is crucial as a contribution to further maize breeding work.
I suggest editing the title of the manuscript to be consistent with the content. eg. Comparison of the nutritional and health-promoting value of grain of new hybrid maize varieties
Please avoid the so-called "mental abbreviations" and not to use the plant name where the grain is mentioned.
When specifying the wavelength (test methods), please complete the a symbol.
Please verify the availability of more recent literary reports (ref. 23-28) and add to the discussion.
The rest of the comments were included in the manuscript file.
Kind regards
Reviewer

Author Response
Reviewer #4:
The topic of comparing the nutritional and pro-health value (based on the content of bioactive ingredients) presented for review is interesting. At the same time, I would like to point out that the discussion of the results requires supplementing the issue of the content of selected bioactive elements, the influence of the genome, and location. This is crucial as a contribution to further maize breeding work.
R= Thank you for your kind comments, we will make the pertinent changes to improve the manuscript.
I suggest editing the title of the manuscript to be consistent with the content. eg. Comparison of the nutritional and health-promoting value of grain of new hybrid maize varieties
R= We have changed the manuscript title to more accurately represent it’s contents.
Please avoid the so-called "mental abbreviations" and not to use the plant name where the grain is mentioned.
R= we have corrected the “mental abbreviations” and avoided the use of the plant name where the grain is mentioned.
When specifying the wavelength (test methods), please complete the a symbol.
R= we have completed the symbols
Please verify the availability of more recent literary reports (ref. 23-28) and add to the discussion.
R= We revised the literature and found that the ones cited are the best for the manuscript
The rest of the comments were included in the manuscript file.
R= We revised and corrected the comments made in the manuscript

Round 2
Reviewer 1 Report
I have read the edited version of the manuscript. Thank you for considering the comments.